# Large Neuroendocrine Neoplasms of the Duodenum: Description of Two Rare Subtypes and Technical Details on Surgical Treatment

**DOI:** 10.3390/diseases12100259

**Published:** 2024-10-18

**Authors:** Giorgio Lucandri, Giulia Fiori, Flaminia Genualdo, Francesco Falbo, Andrea Biancucci, Vito Pende, Paolo Mazzocchi, Massimo Farina, Domenico Campagna, Emanuele Santoro

**Affiliations:** 1Department of Surgery, San Giovanni Addolorata Hospital, 00184 Rome, Italy; giulia.fiori@uniroma1.it (G.F.); flaminia.genualdo@asl.vt.it (F.G.); ffalbo@hsangiovanni.roma.it (F.F.); abiancucci@hsangiovanni.roma.it (A.B.); vpende@hsangiovanni.roma.it (V.P.); pmazzocchi@hsangiovanni.roma.it (P.M.); mfarina@hsangiovanni.roma.it (M.F.); esantoro@hsangiovanni.roma.it (E.S.); 2Department of Pathology, San Giovanni Addolorata Hospital, 00184 Rome, Italy; dcampagna@hsangiovanni.roma.it

**Keywords:** neuroendocrine tumor, gangliocytic paraganglioma, pancreaticoduodenectomy, pancreaticogastrostomy, duodenum, ampullary region

## Abstract

Background: Duodenal neuroendocrine tumors (NETs) are uncommon, accounting for less than 4% of all gastrointestinal neoplasms. Prognosis is related to tumoral staging and grading, as well as to the specific subtype. In this article, we retrospectively describe the clinical presentation and surgical treatment of two rare large duodenal NETs: a high-grade G3 NET and a Gangliocytic Paraganglioma (GP). Methods: Both patients presented with moderate-to-high-degree abdominal pain, without jaundice. Main vessel involvement and metastatic spread were excluded with imaging, while preoperative bioptic diagnosis was obtained via percutaneous needle citology and endoscopic ultrasound. Results: The presence of a sessile large lesion contraindicated any conservative approach in favor of a pancreaticoduodenectomy (PD). The detection of soft pancreatic tissue and a narrowed main pancreatic duct led us to perform a pancreaticogastrostomy to restore proper pancreatic drainage and to minimize the risk of postoperative leakage. Conclusion: PD may be a favorable choice in these cases; this procedure is challenging, but it results in a safer and more favorable clinical outcome for our patients. Pancreaticogastrostomy may guarantee lower rates of postoperative leak and appears to be preferred in this subset of patients.

## 1. Introduction

Neuroendocrine tumors (NETs) represent an unusual subgroup of diseases, usually defined as NENs (neuroendocrine neoplasms). NENs may be classified into two different subgroups: well-differentiated, low-proliferating NENs, called NETs or carcinoids, and poorly differentiated, highly proliferating NENs, called neuroendocrine carcinomas (NECs). NETs can arise almost everywhere in the body: approximately 70% of NETs grow in the gastroenteropancreatic tract (GEP), 25% grow in the respiratory tree, and 5% grow in other sites [1]. NETs in duodenal locations are uncommon, accounting for less than 4% among all gastrointestinal NETs; more than 90% of duodenal NETs arise in the first and second parts of the duodenum and are usually classified as ampullary (within or close to the major or minor papilla) or extra-ampullary [2]. As in other sites, duodenal neuroendocrine tumors (NETs) may be classified according to the presence of hormonal secretion, with about 30% of them being nonfunctional [3]. The World Health Organization (WHO) 2019 NET grading classification is focused on the biological traits of the tumor as they strongly influence prognosis [4]; as the Ki67 index and mitotic count increase, prognosis worsens (NET G3), even in the absence of nodal or liver metastatic disease [5]. Most duodenal NETs (65–80%) are classified as low grade (G1), while high-grade (G3) NETs are an extremely uncommon finding [6]. As well as the biological phenotype, biomarkers like Serum Chromogranin A and Neuron-specific Enolase also appear to be associated with patient prognosis. Upper endoscopy with biopsies and endoscopic ultrasound represent the main tools for the assessment of diagnosis and extent of local spread [4,5]. Nodal metastases are present in 40–60% of cases, while synchronous liver metastases may occur in 10% of patients [7,8]. Another extremely rare subtype of duodenal NET is represented by Gangliocytic Paraganglioma (GP); GP is characterized by triphasic cellular differentiation, epithelioid neuroendocrine cells, spindle cells with Schwann cell differentiation, and ganglion cells [9]. GP usually arises from the second part of the duodenum as a single pedunculated or sessile lesion, and it is associated with a favorable clinical outcome and good long-term survival [10]. GP was first described by Dahl et al. [11] in 1957 and characterized as a benign nonchromaffin paraganglioma by Taylor and Helwig in 1962 [12]. Due to its neuroendocrine component, this tumor is often mistaken for other gastrointestinal tract neoplasms, such as gastrointestinal stromal tumors (GISTs) and NET G1 [13,14]; morphological and immunohistochemical similarities between GP, GIST, and NET G1 may lead to misdiagnosis, but it is important to differentiate GP from other duodenal malignancies as it usually shows a favorable clinical outcome. Preoperative diagnosis is however challenging, as mucosal biopsies are often non-diagnostic because of the submucosal location of the tumor [15,16,17]. Endoscopic ultrasound is a useful tool for diagnosis and lymph node staging, facilitating the choice of proper treatment. Resection of the tumor is the only definitive therapy, and this may be achieved through either endoscopic or surgical resection, depending on the size of the lesion [18,19,20,21]. There are a few cases of metastatic spread of GP to regional lymph nodes [19].

The management of a sessile duodenal NET is often challenging for the clinician. Large size usually contraindicates both endoscopic removal and conservative surgery; on the other hand, pancreaticoduodenectomy (PD) is associated with a certain rate of life-threatening morbidities even in hub centers. The size and location of the tumor make the therapeutic choice extremely difficult. In the study presented herein, we retrospectively report the cases of two patients presenting with large G3 NET and GP duodenal tumors, both successfully treated with PD via an open route and pancreaticogastrostomy. Some technical tips about this kind of treatment of the pancreatic remnant are discussed.

## 2. Case Experience

### 2.1. Case Report 1

A 65-year-old male came to our attention with non-specific abdominal symptoms, asthenia, and weight loss (BMI 18.7), despite normal food intake and regular canalization. Upon physical examination, a palpable mass was found in the right abdomen. His blood count showed moderate anemia (Hb 9.8 g/dL), without any other considerable alteration during blood tests. An Angio-CT scan showed a bulky mass measuring 10 × 9.5 × 7 cm, localized in the upper duodenal-pancreatic area (Figure 1A). Further assessment included an endoscopic biopsy that confirmed the duodenal origin of the neoplasm but was not diagnostic; a magnetic resonance imaging (MRI) scan excluded any possible involvement of surrounding structures and did not show any extraduodenal metastases (Figure 1B). Both Chromogranin A and Neuron-specific Enolase levels increased (respectively 144 ng/mL (nv < 88) and 27.6 ng/mL (nv < 15.2)), pointing the diagnosis toward a duodenal NET. A diagnostic laparoscopy excluded peritoneal seeding of the tumor and allowed us to perform percutaneous biopsies of the duodenal lesion. The histological analysis confirmed the presence of a high-degree NET (G3). Extra-abdominal tumoral localizations were ruled out through a 68-Ga-DOTATOC PET/CT; therefore, our multidisciplinary board authorized direct surgical treatment. The surgical approach was performed with a subcostal bilateral incision; the ascending colon and the root of the transverse mesocolon were exposed and confirmed as not infiltrated; the hepatic pedicle and the mesenterico-portal axis were dissected from the tumoral bulk (Figure 1C), with some difficulties for a 3 cm tract of the superior mesenteric vein, which appeared strongly adherent but not infiltrated by the tumoral tissue. The upper duodenal location and its contiguity to the gastric antrum led us to perform a Whipple procedure. Drainage of the pancreatic remnant was achieved through pancreaticogastrostomy; since the main pancreatic duct was narrowed, and the pancreatic tissue appeared to be too soft to guarantee a safe pancreaticojejunostomy, both biliary and pancreatic anastomoses were stented using a 15 cm 8-Fr Bracci catheter. Intestinal continuity was finally restored with an end-to-side hepaticojejunostomy and an end-to-side gastrojejunostomy. The patient recovered well after surgery (Clavien–Dindo grade 1); the peak of the drainage amylase level was reported on the third postoperative day (725 IU/L), without any sign of clinical activity (Grade A Pancreatic Fistula). The patient was discharged on the thirteenth postoperative day. The pathological study of the lesion showed a rounded and multinodular mass, the cut surface was stiff, fleshy, and pink to grey in color. The definitive histological examination confirmed the bioptic diagnosis of a high-grade duodenal G3 NET, considering the rate of mitotic cells and the presence of necrotic spots. The patient’s Ki67 index was >90%, with more than 20 mitoses per 10 high-power fields. Immunohistochemical panels (BenchMarK Ultra Ventana Roche^©^ Platform) displayed positivity for Cytokeratin 7 (CK7), Chromogranin A (CgA) and Synaptophysin (Figure 1D). All 29 resected perivisceral and regional nodes were found free of disease; thus, the tumor was finally staged as pT3 pN0 pMx G3 R0 (AJCC 8th Ed.). Considering the size and the high histological grade of the lesion, an adjuvant treatment protocol was administered to the patient (first-line cytotoxic systemic chemotherapy, Carboplatin plus Etoposide), without significant side effects; at 15 months after surgical treatment, the patient did not show any sign radiological or clinical recurrence.

### 2.2. Case Report 2

A 56-year-old female presenting with abdominal pain (3.5 points in the Pain Scale Chart), nausea, and constipation in the last 2 months was admitted to our Oncological Surgery Outpatient Service; her nutritional status was good (BMI 23), while her Karnofsky Performance Status (KPS) was scored as 80%. An Angio-CT scan ordered by her general physician revealed the presence of a 5.6 × 4.6 × 5 cm bulk apparently arising from the third duodenal portion, without evidence of any distant metastasis (Figure 2A). The patient did not complain of jaundice nor other signs of biliary obstruction, while at the physical examination, moderate tenderness could be detected in the periumbilical area. Canalization and food intake were described as regular. After prompt hospitalization, the patient underwent an MRI scan which confirmed the duodenal origin of the tumor and the absence of metastasis in the abdomen (Figure 2B). Eco-endoscopic Tru-Cut^®^ biopsies enabled the diagnosis of “Composite gangliocytoneuroma and neuroendocrine tumor (CoGNET) or Gangliocytic Paraganglioma” and the endoscopic examination of the duodenum confirmed its extramucosal origin. The panel of tumoral markers showed a slightly increased value of chromogranin A (95 ng/mL), while neuron-specific enolase was within range. The multidisciplinary tumor board addressed the patient with a direct surgical approach; the large size and sessile shape of the lesion made PD the treatment of choice. After a midline laparotomy, a complete Kocher maneuver and dissection of the hepatic pedicle confirmed the absence of main vessel involvement. The GP presented as an 8 cm duodenal bulk, with a clear sessile structure, arising from the lower duodenal portion (Figure 2C). After a Whipple PD was performed, the detection of a smooth pancreatic remnant and a small main pancreatic duct led us to perform a pancreaticogastrostomy in order to minimize the risk of pancreatic leak; end-to-side hepatico-jejunostomy and end-to-side gastrojejunostomy completed the surgical reconstruction; both pancreatic and biliary anastomoses were stented by Bracci catheters. The postoperative course was uneventful, except for mild pleural effusion not requiring specific treatment (Clavien–Dindo grade 1); the maximum amylase level in the drainage was 430 IU/L on the third postoperative day, without any sign of clinical activity (Grade A Pancreatic Fistula). The patient was discharged on the eleventh postoperative day. Upon macroscopic examination of the resected specimen, an 8 cm sessile duodenal bulk was observed; the tumor showed extra-mucosal localization, while the cut surface presented as whitish to grey; histologic hematoxylin and eosin and immunohistochemical staining (BenchMarK Ultra Ventana Roche© Platform) confirmed the diagnosis of Gangliocytic Paraganglioma (Figure 2D). The tumor was categorized as “composite gangliocytoma/neuroma and neuroendocrine tumor” (Ki67 ˂5%, Chromogranin A+/Synaptophysin+/pS100+/GATA3+); no metastases were found on the 18 resected nodes. Thanks to these favorable prognostic features, the patient underwent a simple follow-up protocol; at 12 months after surgical treatment, there were no clinical or radiological signs of recurrence.

## 3. Discussion

Duodenal neuroendocrine neoplasms are extremely rare, and the international literature usually deals with case reports or small series at best; thus, it appears difficult to state definitive conclusions and landmarks for the proper management of this kind of tumor.

Patients with large and sessile duodenal tumors should be recommended for surgical resection; whenever the radiological preoperative assessment excludes any involvement of the main vessels, PD can be the treatment of choice. As shown in the review by Okubo et al. [19] published in 2016 and dealing with 263 patients operated for large neuroendocrine neoplasms, PD is the most frequently selected surgical procedure, both for NETs and GP (65–80%); such a surgical technique appears mandatory particularly for large or infiltrating tumoral bulky masses, as in both cases reported herein.

Conservative treatment (endoscopic procedures in particular) should be reserved for small, pedunculated neoplasms detected during screening endoscopic procedures, usually measuring less than 2 cm [14,22]. Experience with limited surgical duodenal excision or enucleation as an alternative to PD is uncommon and always associated with conflicting results, considering that a certain rate of lymph node metastases may occur even in small tumors; indeed, only PD allows for extensive lymph node dissection and maximizes the ability to achieve curative resection [23]. Laparoscopic enucleation has been recently reported as the treatment of choice for small low-grade NETs embedded into the duodenopancreatic area [24].

Concerning the surgical procedure, Whipple pancreatoduodenectomy should be preferred over pylorus-preserving techniques in tumors arising from the duodenum as antrectomy is associated with a higher number of resected lymph nodes and a higher rate of disease-free resection margin on the gastric stump [25]. Moreover, patients with duodenal neoplasms do not usually have pancreatic chronic diseases or any kind of biliary obstruction; this means that the pancreatic tissues are usually not sclerotic and the main pancreatic duct is not altered.

Any reconstruction through pancreatico-duodenostomy or Wirsung-jejunostomy should be avoided in these patients because of the high risk of pancreatic fistula. A valuable alternative is represented by reconstruction via pancreaticogastrostomy, which has been associated with a decreased rate of both pancreatic fistula and mortality [26]; the risk of intra-gastric bleeding can be safely managed with the careful hemostasis of the submucosal gastric vessels [27]. In our experience, after proper mobilization of the pancreatic body, its stump is inserted into the posterior aspect of the gastric wall via a 3 cm gastrotomy and fixed using a double 2/0 polypropylene purse-string suture. The pancreatic duct is stented with an 8-Fr Bracci catheter to minimize the risk of postoperative fistula (Figure 3A). Both purse strings should be carefully tightened to ensure proper coalescence of the pancreatic capsule to the gastric parietal layers; it is important to avoid stenosis of the main pancreatic duct, which could cause postoperative pancreatitis (Figure 3B).

Large duodenal NETs may infiltrate surrounding organs (the right colon, gallbladder, and adrenal gland) and resectable liver metastases may be present; in both cases, surgical resection is not contraindicated, although it can be technically challenging [22,28]. The tissue of origin does not seem to influence the prognosis, which depends largely on the staging and grading of the tumor; we can say that despite the histologic diagnosis, the surgical technique is chosen mainly according to the size of the tumor and the depth of infiltration of surrounding structures [22,23,29]. Only a small subset of patients, such as those with purely cystic pancreatic NETs, are always related to favorable pathological and clinical findings (a higher G1 rate and the absence of nodal and distant metastases) [30]. The 5-year disease-free survival rate has been reported to be as high as 85–95% for early-stage disease, 65–75% for locally advanced disease, and 20–40% for patients with liver or distant metastases at the time of first diagnosis [3].

In the first patient, the presence of a nonfunctional G3-NET with a Ki67 index > 90% led us to suggest early adjuvant treatment with systemic chemotherapy and a close follow-up regimen; in the second patient, the presence of GP without metastasis and a Ki67 index ˂ 5% indicated a simple follow-up regimen after surgery.

## 4. Conclusions

In conclusion, we strongly support Whipple PD as the treatment of choice in large neuroendocrine duodenal tumors; pancreaticogastrostomy is a simple and reliable method for the management of the pancreatic remnant.

## Figures and Tables

**Figure 1 diseases-12-00259-f001:**
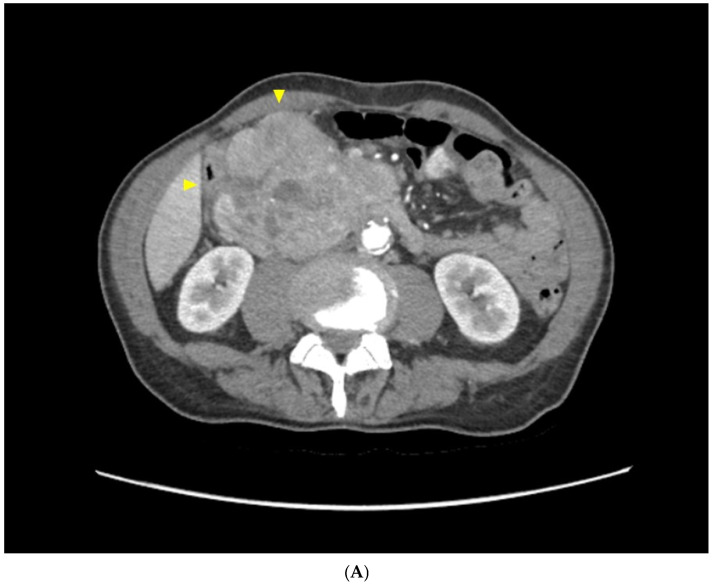
(**A**) Contrast CT scan: bulky mass with inhomogeneous uptake, arising from the upper duodenal–pancreatic angle. (**B**) The MRI main lesion appears hypointense on fat-suppressed T1-weighted sequences. (**C**) Intraoperative appearance of the surgical bed after PD: a 3 cm tract of the superior mesenteric vein appears strongly dissected but not injured because of retromesenteric tissue removal (arrowheads). (**D**) Histopathological evaluation (**A**) with hematoxylin and eosin staining of the mass (10× magnification). (**B**) IHC slide showing diffuse positivity for chromogranin A (40× magnification).

**Figure 2 diseases-12-00259-f002:**
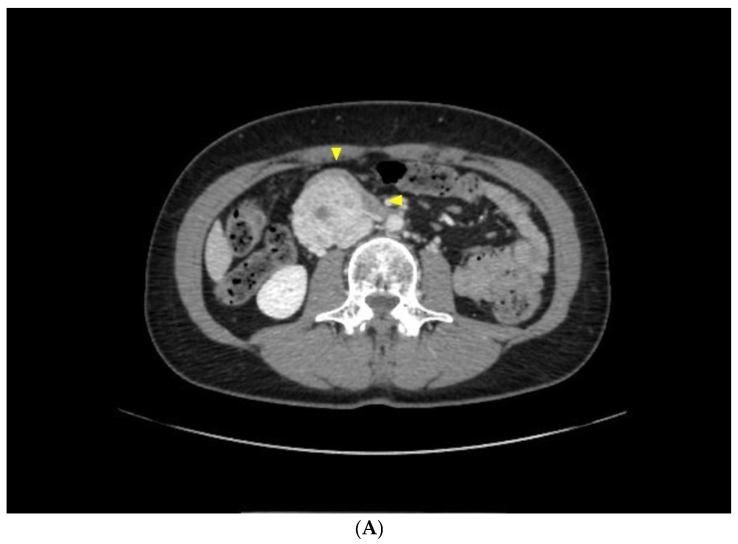
(**A**) The Angio-CT scan revealed a 5.6 × 4.6 × 5 cm bulk apparently arising from the third duodenal portion, without evidence of metastasis (arrowheads). (**B**) MRI scan confirming the localization of the tumor (arrowheads) and the absence of other bulk masses in the abdomen. (**C**) Intraoperative aspect of the duodenal lesion; resected specimen from the Whipple pancreatoduodenectomy (PD). D, duodenum; T, tumor. (**D**) Histopathological evaluation (**A**) with an IHC slide showing diffuse positivity for chromogranin A. (**B**) Nuclear and cytoplasmic staining in tumor cells with S-100 protein. (**C**) Hematoxylin and eosin staining.

**Figure 3 diseases-12-00259-f003:**
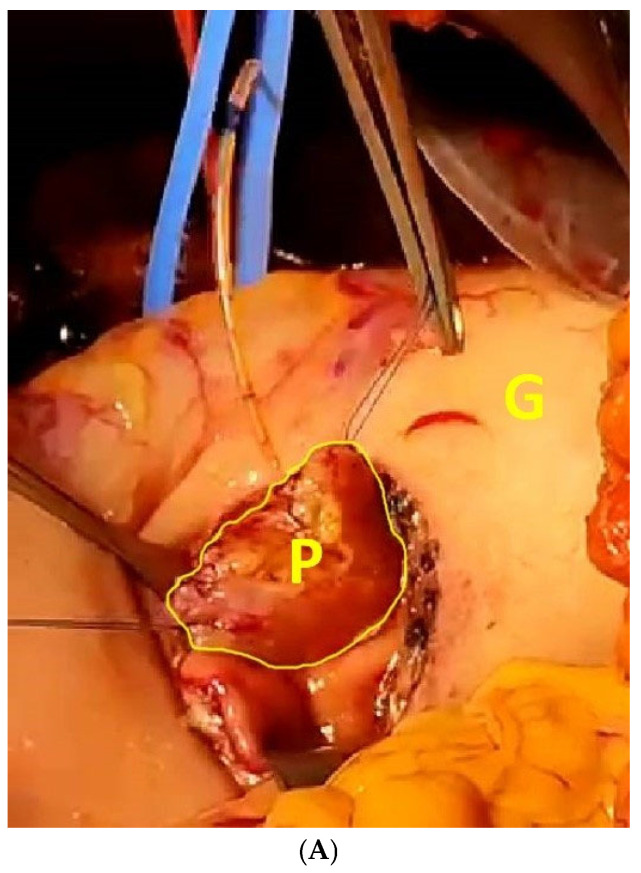
(**A**) Intraoperative aspect of the pancreaticogastrostomy; the pancreatic remnant (P) has yet to be introduced into the gastric lumen (G) during a 3 cm posterior gastrotomy. The main pancreatic duct has been stented with an 8 Fr Bracci catheter. (**B**) Detailed appearance of the pancreaticogastrostomy, showing both polypropylene purse string sutures not yet tightened around the pancreatic body (arrowheads). G, gallbladder; P, pancreas.

## Data Availability

The raw data supporting the conclusions of this article will be made available by the authors on request.

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
