# Peer review of "Large Neuroendocrine Neoplasms of the Duodenum: Description of Two Rare Subtypes and Technical Details on Surgical Treatment"

_diseases, 2024, doi:10.3390/diseases12100259_

Round 1
Reviewer 1 Report
Comments and Suggestions for Authors
Congratulations to the authors for drawing attention to an uncommon but technically challenging primary site of neuroendocrine neoplasms. These two cases certainly represent a rare circumstance that would be a helpful addition to the neuroendocrine literature.
However, concerns exist regarding the overall message of this report. If the authors' main point is that bulky tumors of the duodenum may be surgically resected via a pancreaticoduodenectomy, as is implied by its prominence within the conclusions section, it is unclear that this adds to the understanding of the surgical management of oncologic duodenal resection. Additionally, pancreaticogastrostomy reconstruction for patients with high risk of leak would not be a novel approach and is well supported by the existing literature, and both cases represent patients that still had Grade A pancreatic fistulae despite this reconstruction method. One point of significant interest in the field would the successful surgical management of a high grade neuroendocrine neoplasm with aggressive surgical resection that resulted in no recurrence at 15 months; it is my recommendation that the authors reformat the case report to focus on this management strategy and report the first case only, in combination with a thorough literature review of management in high grade NENs. Regardless, the authors' recommendation that duodenal NENs be approached using PD is not supported by the use of a case report format and should be revised to use language that accepts that this report does not represent a thorough overview of all situations.
Specific feedback:
For case 1, please include the factors that led to choice of a classic Whipple procedure for operative resection, as it was described as an intraoperative decision and intraoperative findings are implied to be the reason this choice was made.
Was the tumor determined to be well-differentiated or poorly differentiated? The choice of carboplatin plus etoposide for adjuvant chemotherapy would be most commonly employed in a case of poorly differentiated neuroendocrine carcinoma; please discuss the factors that were used to make this decision and address that if this were known to be a poorly differentiated duodenal neuroendocrine carcinoma preoperatively, aggressive operative resection would be a more controversial approach. However, if the point of reporting this case was to emphasize that a poorly differentiated neuroendocrine carcinoma of the duodenum may be effectively treated with aggressive oncologic resection and adjuvant chemotherapy with acceptable morbidity and excellent oncologic outcomes, please ensure that that is clarified to the reader.
Comments on the Quality of English LanguageAlthough the report flows well and the authors are clearly knowledgeable regarding their subject, there are number of small grammatical issues that should be addressed:
Line 36 - should be worded 90% of duodenal NETs rather than gastrointestinal NETs
Line 38 - avoid capitalization for the word neuroendocrine
Line 44 - Classified rather than classifies
Line 57 - Gastrointestinal does not require hyphenation
"Related with" often used when the phrase "associated with" should be chosen instead
Please clarify line 96 "the mesenterico-portal axis were dissected from the tumoral bulk (Fig. 1C), even if a 3 cm tract of superior mesenteric vein appeared strictly surrounded but not infiltrated by duodenal tumor."
Author Response
- We thank You for Your very interesting and remarkable comments; they led us to further reflections and surely will contribute to improve the quality of our manuscript.
- We agree with Your statement that the most appropriate treatment for large duodenal tumors is PD, because no alternatives can be considered. We simply pointed out that the most appropriate reconstructing procedure after PD is the Whipple one, because antrectomy may guarantee appropriate resection margin in case of upper duodenal lesion and larger nodal dissection.
- Moreover, we consider that a smooth pancreatic remnant can be better treated by pancreato-gastric reconstruction. In these cases, operative strategy is always discussed and planned preoperatively, even if the ability of introducing some i.o. changes to what we have planned before can be considered part of our job. Some technical tricks discussed about this reconstruction (double purse string suture i.e.) represent the results of our experience.
- Consider that Grade A Pancreatic fistula is defined only by serum levels, without any clinical consequences.
- Any decision about therapeutical aspects is always discussed with our oncological equipe: in our case report 1, presence of G3 duodenal NET, without nodal metastases but with large loco-regional extension (adhesion but no parietal involvement of superior mesenteric vein) led us to an aggressive approach, both for the extent of resection and for adjuvant treatment.
- Please clarify line 96 "the mesenterico-portal axis were dissected from the tumoral bulk (Fig. 1C), with some difficulties for a 3 cm tract of superior mesenteric vein, which appeared particularly adherent to the tumoral tissue."
- According to Your suggestions, language modifications have been introduced and the entire text has been further reviewed by a native English speaker.

Reviewer 2 Report
Comments and Suggestions for Authors
Our research colleagues diagnosed two large duodenal neuroendocrine neoplasms; histology then allowed the histological identification of a NET and a gangliocytic paraganglioma. Unfortunately, neuroendocrine tumors are constantly increasing, and if until 15 years ago they were diagnosed from the age of sixty onwards, today we also find them in younger patients. The process includes diagnosis, surgical treatment where possible, and adjuvant chemotherapy where indicated by histological examination and imaging.
In the introduction to the clinical cases it is advisable to point out that imaging is fundamental for hypothesizing the diagnosis, given that the images of the neoplasm are quite indicative with the peripheral contrast and the sort of more central shadow, which distinguishes these pathologies. Furthermore, the CT scan allows us to make a further diagnostic and therapeutic judgment, allowing us to establish the degree of progression of the disease with metastases and lymph nodes where the pathology is located. Other diagnostic aids can include MRI with a tracer containing octeotride if the cells are sufficiently differentiated and have receptors on their surface. Otherwise, the PET scan with GALLIUM is fundamental as it allows you to identify the primary lesion and any secondaries. If there are no metastases, the treatment is certainly surgical first and the adjuvant therapy still has somatostatin or similar as its backbone, keeping in mind the potential risks on the gallbladder (PMID: 38051513 to be cited in the bibliography) everolimus (DOI: 10.1016 /j.trre.2015.09.001) also important for the inhibition of calcineurin. Oncology immunotherapy, which is increasingly being put into practice, should be taken into consideration. The description of the cases is well conducted, the emphasis on the discussion of the case in the multidisciplinary commission is excellent and the right emphasis is placed on endoscopy. He wonders why his colleagues preferred an external puncture rather than through the bowel through which the neoplastic mass was easily reachable. We absolutely agree on pancreaticoduodenectomy for both cases. It was the only reliable therapy. It is also advisable to say a few words about the follow-up since the tumors were important for the degree of differentiation and size. Excellent iconography with histological preparations, English needs improvement. The bibliography is the basis on which the entire work is built
Comments on the Quality of English Languageenglish needs to be revised
Author Response
- We thank You for Your very interesting and remarkable comments; they led us to further reflections and surely will contribute to improve the quality of our manuscript.
- We really appreciated your focus on the importance of imaging in the diagnosis of these rare forms of duodenal tumors; the importance of both CT scan and NMR has been highlighted in the introduction, as well as the role of PET scan in excluding distant metastases.
- Please consider that also Demographic, Clinical, Therapeutical and Prognostic aspects should be emphasized at the same time, respecting the length of the entire manuscript as requested for case report by Curr Onc.
- External puncture of the bulk has been performed during preoperative laparoscopy, after endoscopic biopsies resulted as non-diagnostic.
- According to Your request, a further revision of the quality of English language has been conducted by a native English speaker.

Reviewer 3 Report
Comments and Suggestions for Authors
Lucandri G et al. diligently reviewed two rare subtypes of large neuroendocrine neoplasms of the duodenum with well-addressing the technical details of surgical of treatment. The authors should be commended for such as a surgical endeavor because this article will be a good teaching material for young surgeons.
After the perusal of the whole manuscript, I find the manuscript was well written, clinically very important for the clinician to treat the GI tract for NET.
However, to facilitate the clinical comprehensiveness of this article, I would like to suggest the authors to discuss further on the current clinical work regarding the histopathologic comparison between duodenal and pancreatic NETs (1. Asian Journal of Surgery. Volume 46, Issue 2, February 2023, Pages 774-779; 2. Asian Journal of Surgery. Volume 45, Issue 12, December 2022, Pages 2659-2663), establishing the prognostic nomogram for patients (Asian Journal of Surgery. Volume 47, Issue 1, January 2024, Pages 433-442) and even the technical feasibility of minimally invasive enucleation (Wu J et al. Laparoscopic enucleation of tumors embedded in the pancreatic head: Safety and feasibility. Asian Journal of Surgery Available online 12 September 2024), which have been mentioned before from the Asian fellow researchers.
Comments on the Quality of English LanguageQuite good.
Author Response
We thank the reviewer for his positive judgement on our manuscript.
Your suggestions gave a significative contribute to improving its contents and comprehensiveness of our aims.
Two articles You suggested (taken from Asian Journal of Surgery) have been added on discussion and bibliography.
We thank You once again, hoping for further cooperation.

Reviewer 4 Report
Comments and Suggestions for Authors
The use of English language is very poor that it should be improved.
The authors present two cases of very large neuroendocrine tumors of duodenum that were managed by Duodenal and pancreatic head resection.
They explain the diagnostic work-up and surgical procedure in detail.
This is an informative study. Well designed and presented.
Comments on the Quality of English LanguageIt should be extensively revised by a professional agency.
Author Response
We thank the reviewer for his positive judgment on our manuscript.
Following Your suggestion, text has been reviewed and corrected for a second time by a native english speaker; some corrections has been highlighed as requested by editorial rules.
We thank You once again, hoping for further cooperation.

Round 2
Reviewer 1 Report
Comments and Suggestions for Authors
Significant improvements have been made to the phrasing and conclusions of this case report.
Some minor edits are still required. Please remove any contracted words ("don't") and replace them with the written out form ("do not").
Also, the abstract should be adjusted to reflect the revised conclusions. Specifically, the abstract currently states: "Large duodenal NETs should always be treated with PD; this procedure, even if challenging, should be preferred to partial resection and enucleation, as it results in a safer and more favorable clinical outcome." The case report format of this study does not support a sweeping recommendation as is stated here, and it should be adjusted to reflect that the these cases support that PD may be a favorable choice, but prospective study would be necessary to support this recommendation.
Author Response
Dear reviewer, we really appreciated Your positive judgment about our manuscript and we are going to modify it according to Your suggestions.
Any contracted word has been removed and replaced in the text
Abstract has been modified, supporting PD as the gold standard treatment for such kind of tumors. Please consider that large duodenal neuroendocrine tumors are extremely rare and these lesions usually arise from pancreatic head. This make a prospective collection very difficult; however we support Your statement, and maybe creation of a multicenter international registry on duodenal NET could be welcome. Thank You for Your suggestions once again.

Reviewer 2 Report
Comments and Suggestions for Authors
The few changes made to the paper make it more precise and easy to read. We don't need to add much more, it's a good case report whose publication is encouraged. The photographs are excellent, English is very good. The bibliography represents a good support on which the paper is built.
Author Response
Dear reviewer, we really appreciated Your positive judgment about our manuscript. We hope for further collaboration and thank You once again.
